# Acoustic Sensing Fiber Coupled with Highly Magnetostrictive Ribbon for Small-Scale Magnetic-Field Detection

**DOI:** 10.3390/s25030841

**Published:** 2025-01-30

**Authors:** Zach Dejneka, Daniel Homa, Logan Theis, Anbo Wang, Gary Pickrell

**Affiliations:** 1Department of Materials Science and Engineering, Virginia Tech, Blacksburg, VA 24061, USA; dan24@vt.edu (D.H.); pickrell@vt.edu (G.P.); 2Center for Photonics Technology, Bradley Department of Electrical and Computer Engineering, Virginia Tech, Blacksburg, VA 24061, USA; lbtheis@vt.edu (L.T.); awang@vt.edu (A.W.)

**Keywords:** optic sensors, magnetostriction, magnetism, magnetic-field sensors, distributed acoustic sensors

## Abstract

Fiber-optic sensing has shown promising development for use in detecting magnetic fields for downhole and biomedical applications. Coupling existing fiber-based strain sensors with highly magnetostrictive materials allows for a new method of magnetic characterization capable of distributed and high-sensitivity field measurements. This study investigates the strain response of the highly magnetostrictive alloys Metglas^®^ 2605SC and Vitrovac^®^ 7600 T70 using Fiber Bragg Grating (FBG) acoustic sensors and an applied AC magnetic field. Sentek Instrument’s picoDAS interrogated the distributed FBG sensors set atop a ribbon of magnetostrictive material, and the corresponding strain response transferred to the fiber was analyzed. Using the Vitrovac^®^ ribbon, a minimal detectable field amplitude of 60 nT was achieved. Using Metglas^®^, an even better sensitivity was demonstrated, where detected field amplitudes as low as 3 nT were measured via the strain response imparted to the FBG sensor. Distributed FBG sensors are readily available commercially, easily integrated into existing interrogation systems, and require no bonding to the magnetostrictive material for field detection. The simple sensor configuration with nanotesla-level sensitivity lends itself as a promising means of magnetic characterization and demonstrates the potential of fiber-optic acoustic sensors for distributed measurements.

## 1. Introduction

Magnetic sensors are used in many different applications. In patient care services, diagnostic tools and sensors are crucial for quickly and efficiently assessing conditions or problems in a patient that can make the difference for a successful recovery. Cardiovascular and neurological activity consist of very small electrical and magnetic signals that can be monitored to determine heart and brain health or recognize underlying conditions [1]. Traditional monitoring in the heart revolves around electrocardiograms (ECGs) for characterizing the cardiovascular condition of a patient. However, they are difficult to use for continuous measurements, as adhesive pads need to be applied, which can cause skin irritation. Patients also need to have an adequate surface for the pads to be applied, which frequently involves quick hair removal. Electroencephalograms (EEGs) used for brain activity operate similarly in that electrodes need to be attached to the skin. Obtaining cardiovascular and neurological signals for continuous monitoring, thus, can be difficult or uncomfortable for the patient [2,3]. Because of this, advances have been made in magnetic sensors for non-contact use in patient heart and brain measurements. However, devices and sensors in the biomedical field used for real-time magnetic sensing diagnostics are expensive or can be invasive in other ways. Often, biomedical procedures involving magnetic-field measurements depend on injected nanoparticles to produce “targets” that can be reliably tracked [1]. Other procedures such as magnetocardiography (MCG) and magnetoencephalography (MEG), while not invasive, rely on superconducting quantum interference devices (SQUIDs) that need to operate at cryogenic temperatures and are very expensive to produce even though they provide reliable low-field detection [2,4]. Magnetoimpedance (MI)-based sensors are another method for highly sensitive magnetic-field measurements and have more recently attracted interest in health monitoring with MCG and MEG. Able to resolve field amplitudes in the pico-tesla regime as well as not having to rely on cryotemperatures to operate, the technique has good potential. However, MI is still based on point measurements and does not have distributed sensing capabilities [5]. Noninvasive and distributed substitute devices for the magnetic sensing of heart and brain signals would be a monumental leap in the healthcare industry.

Downhole applications in the energy industry also must rely on magnetic sensors such as fluxgate magnetometers (FGMs) or surface nuclear magnetic resonance (surface-NMR) for monitoring. FGMs are crucial for measuring solar activity as well as for geophysical monitoring [6]. Cross-well imaging and directional drilling use FGMs as acting magnetic-field point sensors. This makes the entire process of taking measurements tedious, as the magnetic-field sensor is only one dimensional and may need to be deployed repeatedly to accurately assess the subsurface conditions. The demand for alternative and cheaper magnetic sensing methods has also drastically increased in recent years, in part due to the reliance of legacy ring cores in fluxgate magnetometer (FGM) production. The legacy cores used in the magnetometers stopped being produced almost three decades ago, and the design was never available to the public due to military involvement further driving up their cost [7]. Surface-NMRs, conversely, can take bulk measurements in real time and are not limited by the scarcity of legacy cores. In some recent studies, NMR has shown newfound potential with advancements in sensing information [8]. However, vertical resolution with NMR is usually limited, and obtaining enough information can be slow and cumbersome [6,8].

Acoustic sensing based on optical fiber is a well-established technology and easily integrated into existing systems. The optical fiber in these systems is robust, flexible, trivial in size, and can be used for distributed measurements. It can also withstand harsh conditions such as extreme temperatures, corrosion, and electromagnetic interference (EMI) [9]. Distributed acoustic sensing (DAS), in particular, has been reliably used as a measuring device for surveillance and structural health monitoring. DAS optical fiber can distinguish acoustic signals across several kilometers along its length [10,11]. The high sampling rate (thousands of Hz) allows for a large amount of information to be obtained about individual vibrations anywhere along the length of the fiber, including the phase, frequency, and amplitude. Used in vehicle monitoring, seismic-activity detection, and various other applications, the distributed nature of the technology and its high sensitivity make it one of the most promising methods for the detection of acoustic signals [11].

The work presented in this paper looks at combining the existing DAS technology with highly magnetostrictive amorphous metal alloys like Metglas^®^ 2605C and Vitrovac^®^ to produce a magnetic sensor alternative to fluxgate magnetometers and other commercialized devices capable of distributed sensing. Developing a magnetic sensor with the physical footprint of just a singular cable and interrogation system (that can be kilometers away from the far end of the cable) while replicating similar levels of commercial sensitivity is significant. Additionally, being able to take advantage of DAS removes the need for repeated deployment of point sensors [12].

In the past, our research group has demonstrated success with fiber-based magnetic-field sensors. The multi-material fibers use in-cladding magnetostrictive wires that are created during the fiber draw process. The magnetostrictive materials tested include nickel, Galfenol, Vitrovac^®^, and Metglas^®^. The various materials provide templates for different effective sensitivities, and because they are completely contained within the 125 μm diameter fiber, the design is elegant and compact for a magnetic-field sensor. However, the best sensitivity of these sensors demonstrated so far was 500 nT using a Metglas^®^ in-cladding wire [13,14]. Additionally, the draw process is arduous, and mismatches in the thermal expansion coefficients of the fused silica glass and magnetostrictive metal lead to various difficulties. The size of the in-cladding metal wires also makes fusion splicing difficult and, in some cases, impossible for proper integration in fiber systems. A simpler approach in this study is taken to improve the sensing resolution while still using DAS technology and highly magnetostrictive materials. The response of a magnetic sensor configuration consisting of a *picoDAS* acoustic sensing fiber simply laid onto an unbonded ribbon of amorphous magnetostrictive metal (Metglas^®^ and Vitrovac^®^) was evaluated.

## 2. Materials and Methods

A 120 m long spool of Sentek Instrument acoustic sensing single-mode fiber (SMF) was used with broadband Fiber Bragg Gratings (FBGs) inscribed every 2 m along the entire length of the fiber. The fiber is commercially drawn at Sentek with gratings containing a periodic index variation. The fiber is designed for 1550 nm wavelength transmitted light. At each grating, which acts as a reflection point, the light is transmitted back through the fiber. The acoustic sensing fiber is then able to take distributed measurements as grating pairs function as Fabry–Perot interferometers. The acoustic sensing SMF was then fusion spliced to a fiber connector to be integrated with Sentek Instrument’s *picoDAS* Interrogator. The interrogator samples at a frequency of roughly 37 kHz and can give strain measurements from the optical fiber sensor with ultra-high sensitivity at less than 0.25 nano strain (nε).

Paired with the Sentek drawn fiber (individually) are two 25 mm wide amorphous metal ribbons, Metglas^®^ 2605SC (Fe_81_B_13.5_Si_3.5_C_2_) and Vitrovac^®^ (7600 T70) Fe_65_Co_18_Mo_0.3_Si_0.8_B_15.5_C_0.4_. The Metglas^®^ measured 0.025 mm in thickness, while the Vitrovac^®^ was 0.021 mm thick. Vitrovac^®^ 7600 T70 has a saturation magnetostriction coefficient of around 47 ppm and Metglas^®^ 2605 SC of 27 ppm. Both metal alloys are designed to be magnetically soft with a high elastic modulus, making them ideal candidates for detectable strain responses at extremely low magnetic-field strengths (highly magnetostrictive) [15,16]. Magnetostrictive materials contain internal magnetic domains (M) that are settled in an equilibrium position dependent on the strain inside a material lattice resulting from the anisotropy energy [11]. When an external magnetic field is applied to the material lattice, these magnetic domains will begin to align along the axis of the applied magnetic field [17,18]. This domain movement, in turn, stresses the material, and a resultant strain is produced, as shown in Figure 1, where λ is the maximum strain response at saturation with a horizontally incident magnetic field.

Upon exposure to an alternating or AC-based magnetic field, the same phenomena occur, but the strain response now also shares a similar periodicity to the input signal. This sinusoidal resultant strain response in the magnetostrictive material produces a vibration. This acoustic signal, derived from an AC magnetic field, is then transmitted through the metal ribbon. When the acoustic sensing fiber is placed on top of the stimulated respective magnetostrictive ribbon, the strain induced from the vibration onto the fiber is detected and then converted to a digital value by the *picoDAS* Interrogator. The system creates a matrix of time, distance, and strain, all gathered from the broadband FBG pairs along the fiber.

An external alternating magnetic field is generated by an air-core solenoid. The solenoid is connected to a signal generator and voltage amplifier, though, at lower magnetic fields for testing, just the signal generator is used. The air-core solenoid measures 2.2 m in length to completely contain the FBG spacing for the Fabry–Perot interferometer pairs and avoid most of the magnetic field non-uniformities caused by the end effects of the solenoid. Inside the solenoid, a 2 m long strip of amorphous metal ribbon is centered with the SMF laid laterally directly on top, as shown in Figure 2.

An alternating current is then applied to the solenoid at a chosen frequency with a sinusoidal wave input. A magnetometer is used to record the generated magnetic-field amplitudes, and then the corresponding response of the sensor configuration is evaluated by the fiber and interrogation system. In the time domain, the Sentek *picoDAS* system measures the dynamic strain created by the magnetostriction in the amorphous metal ribbon. A MATLAB script takes the generated matrix of strain, position, and time to produce an intensity spectrum. The script takes advantage of a fast Fourier transform (FFT) to create the spectrum, looking at a frequency range of up to 20 kHz at the specified position (in this case, along the 2 m span inside the solenoid). The amplitudes of the sinusoidal constituents give the intensity for a respective frequency.

## 3. Results

In this study, two different materials in tandem with the acoustic sensing fiber are investigated to evaluate their respective sensitivity to AC magnetic fields. Different frequencies, various field strengths, and multiple sensors at once were all tested to evaluate the potential of an acoustic sensing fiber and magnetostrictive-ribbon-based sensor without the need for any form of physical bonding. After the ribbon-and-fiber sensor is set up within the air-core solenoid, an AC current is applied, and the frequency response of the fiber is denoted. A frequency spectrum sample is shown in Figure 3. P1(f) indicates the FFT-derived amplitude intensity for a given sinusoidal constituent corresponding with the denoted frequency spectrum along the *x*-axis.

The applied magnetic-field frequency is 350 Hz, and the resultant amplitude spectrum intensity peaks (one-order-of-magnitude difference) are noted at the same frequency. In Figure 3a, the *y*-axis scale is more limited due to the smaller intensity of the 350 Hz peak, and low-frequency spectra responses are more apparent. Because the fiber is so sensitive to acoustic signals, the noise floor is much higher toward the low-frequency end of the spectrum due to factors in the surrounding environment. However, this can later be filtered out in post-processing. This spectrum generation and peak finding is repeated across multiple magnetic-field amplitudes for a given frequency to create a calibration curve for the sensor. The peaks are selected from these generated frequency spectrum intensity plots, like in Figure 3.

The Vitrovac^®^ ribbon was first tested using this method at 100 Hz, and the intensity response, P(f), compared to the magnetic-field amplitude was plotted as shown in Figure 4.

The Vitrovac^®^ coupled with the acoustic sensing fiber was able to detect fields as low as 60 nT and saw an increasing strain intensity response, with growing magnetic-field amplitudes matching the driving frequency of the magnetic field. However, magnetostriction-induced strain in relation to AC magnetic fields usually sees a doubling effect. This means that if an applied magnetic field to the material had a driving frequency of 100 Hz, the main response would be expected at 200 Hz. This is because the magnetostriction curve for a given material (strain vs. field strength) is symmetrical across the *y*-axis and roughly assumes the form of a quadratic fit at low fields [19]. A sinusoidal signal at both its maximum and minimum would, thus, have exactly the same resultant strain response, allowing for the doubling effect [19,20]. However, because the experimental setup is not magnetically shielded in any way, the solenoid and also the sensor are susceptible to other external magnetic fields. The magnetic field of the Earth on the surface is on the order of 25–65 μT. As this is significantly stronger than the field amplitude tested in this experiment, the sinusoidal signal is biased to the point that it never crosses the *y*-axis or the magnetic field strength can be negative. Thus, the resultant strain response remains analogous to the driving frequency of the magnetic field from the solenoid and signal generator as opposed to showing the characteristic doubling in unbiased magnetostrictive materials. The lack of magnetic shielding could also be why there appears to be an error in the response from the Vitrovac^®^. Inconsistencies in the AC voltage signal could also alter the response.

To demonstrate the distributed nature of the sensor, the experiment was repeated but at varied magnetic-field strengths and frequencies and across multiple FBGs. Three air-core solenoids were used, each containing a 2 m length of Vitrovac^®^. The acoustic sensing fiber was snaked through all three solenoids such that the 2 m gauge length of the FBGs was centered with the ends of the solenoids. The solenoid setup is shown in Figure 5.

All three solenoids were magnetically characterized to determine the field amplitudes generated by each one for a given applied voltage and frequency. After the fiber was properly set in the solenoids, various magnetic fields were generated and independently resolved using the acoustic sensing fiber and taking the FFT at each respective position. Figure 6 shows the visual representation from Sentek Instrument’s software analyzing the output of the interrogator.

The sinusoidal response shown in red is the resultant strain from a 100 Hz driving frequency, the pink from 250 Hz, and the white from 500 Hz. All three signals were unique, and the spectrum peaks corresponded to their respective solenoid driving frequency with components of both the doubled response and original sinusoid. The FBG pair in the first solenoid is 4 m away from the FBG pair in the second, which is then 6 m away from the pair in the third and final solenoid. Even though the experiment demonstrates a sensing spatial resolution of 2 m along only 16 m of fiber, the used DAS system is capable of distributed measurements along tens of kilometers of fiber with minimal loss. General DAS fibers typically exhibit 0.4 db per kilometer of loss and have been shown to still resolve over 100 km worth of fiber [11]. While the testing configuration using Vitrovac^®^ ribbon demonstrates the potential of an acoustic fiber and magnetostrictive-ribbon-based distributed sensor, the sensitivity only demonstrates a level of 60 nT, and compared with the fitted quadratic curve, the measured spectra intensities show some discrepancy. Metglas^®^ was next evaluated to compare its strain response to that of the Vitrovac^®^.

The experimental setup used is identical to the one shown in Figure 3, though with the Metglas^®^ 2605SC ribbon used instead. After initial testing with the new alloy ribbon, Metglas^®^ showed more promise in terms of the lowest detectable field as well as consistency and curve fitting. Figure 7 shows a calibration curve of the strain amplitude intensity vs. magnetic field amplitude across a wider range of measured field amplitudes.

A driving frequency of 100 Hz is applied to the air-core solenoid containing the Metglas^®^ ribbon. In this case, however, because the field amplitude is much higher (up to 20 μT), the doubled magnetostrictive response is recorded, as the biasing due to external static fields is less prominent. The correlation of intensity response to magnetic-field amplitude is much smoother. The signal generator and solenoid used for the AC magnetic field could only achieve a minimum amplitude of around 100 nT. To generate alternating signal strengths below this, a 10 dB attenuator had to be added to the setup to reduce the effective current from the signal generator. The solenoid was then recharacterized with the magnetometer. Below 10 nT, the magnetic field strength had to be interpolated using measured current values with a multimeter instead, as the magnetometer resolution was no longer adequate to effectively measure the magnetic-field amplitude. Figure 8 shows the spectrum intensities of a 350 Hz driving frequency and the magnetic field range tested.

Figure 8a is a magnified portion of the plot in Figure 8b to show the lower end of the field amplitude range evaluated. The lowest magnetic-field strength with a resolvable strain intensity peak above the noise floor was 2.74 nT. The curve across the tested range shows a strong positive correlation with strain peak intensities in relation to magnetic-field amplitude.

Both the Vitrovac^®^ and Metglas^®^ sensing configurations were tested at various frequencies and showed similar trending responses. The magnetometer that was used to characterize the solenoids is frequency dependent, and to keep the study concise and clear, only the responses at driving frequencies of 100 Hz and 350 Hz are shown. For the latter, 350 Hz showed characteristics of a resonant response (similar heightened intensities were observed at 175 Hz as well) and was, thus, used to analyze lower field spectrum intensities. While the strain responses across both the Metglas^®^ and Vitrovac^®^ demonstrated consistent positive correlations, there remains deviation within the response quadratic fit. Because the test setup is not magnetically shielded, any external magnetic fields, such as from electronic devices in the room, wiring in the building, or otherwise, could skew the strain response from the amorphous magnetostrictive ribbons. Vibration from the metal itself or environmental movement could also cause the acoustic sensing fiber to shift its position from its centered alignment along the ribbon, which could, in turn, alter the detected strain response by the fiber. However, the fiber was later taped down using Scotch^®^ tape to remove the possibility of the fiber position changing, and no significant difference was shown in sensitivity, but the consistency of repeated tests was improved.

At higher magnetic-field amplitudes, the fit, as shown in Figure 7, works very well in characterizing the magnetostrictive strain response with little deviation. This implies that with a reduction in the noise floor at low field amplitudes, the consistency of curve fitting and field detection will further improve. Additional research and experimentation need to be conducted to improve the consistency and to better characterize the sensor configurations. Adding Mu metal surrounding the solenoid would help limit the external magnetic fields influencing the response in the sensor. Thermal magnetic annealing has been shown to allow for higher strain responses in magnetostrictive materials in other studies [12,16]. Manipulating the amorphous ribbon via thermal annealing could help improve sensitivity. Advances in the signal processing of the data outside of applying only an FFT, such as filtering out low-frequency noise or focusing on just the desired frequency, could also further improve the overall performance of the sensor. In applications where a 2 m gauge length and sensing resolution is excessive, the fiber and ribbon could be coiled to reduce the physical footprint of the sensor with further testing to characterize this. However, for the scope of demonstrating the potential of a simple fiber-based distributed sensor design with the ability to resolve nanotesla-level magnetic-field amplitudes, the results show strong promise for future work and development.

## 4. Conclusions

In this study, a unique sensor design for the detection of small-scale magnetic-field amplitudes is demonstrated using two different magnetostrictive amorphous alloys. In past experiments, magnetic sensing multi-material optical fiber, also using Metglas^®^, showed promise for downhole applications with a minimum sensitivity around 500 nT. However, this is still orders of magnitude above the resolution needed for use in some applications such as cardiovascular studies and patient care. The new design, using an unbonded DAS single-mode fiber with Metglas^®^ ribbon, was able to detect a minimal field amplitude of 2.74 nT, resulting in over two orders of magnitude of improvement. While the sensor is not quite at the needed levels of field detection or consistency to be seen as an effective substitute for SQUIDs, the method demonstrates strong potential for further development and future use in these fields. With the rising costs of fluxgate magnetometers and the limited resolvable information from surface-NMRs, the sensor configuration with nanotesla-level sensitivity, distributed sensing capabilities, and compact design also shows promise for use in energy fields and downhole applications. Optical acoustic sensing fiber is easy and cheap to mass produce for kilometers as well as being very robust. Metglas^®^ can also be manufactured for long lengths and is already commercialized. With future research looking at annealing, signal processing, and cable formulation, in addition to already-promising results as a magnetic sensor, the DAS fiber and magnetostrictive ribbon design demonstrates strong potential for use in many applications including within the energy industry and in the biomedical field.

## Figures and Tables

**Figure 1 sensors-25-00841-f001:**
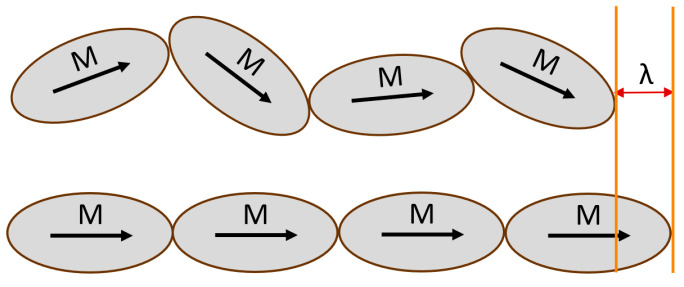
Magnetic domains in a magnetostrictive material lattice at equilibrium (top) and under an external magnetic field at saturation strength (bottom) with a maximum magnetostrictive strain λ.

**Figure 2 sensors-25-00841-f002:**
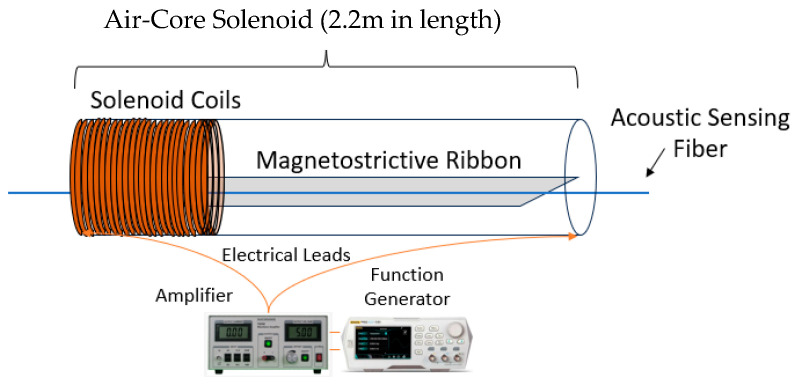
Experimental test setup with air-core solenoid connected to function generator and amplifier. The magnetostrictive ribbon is positioned in the middle of the solenoid with the sensing fiber on top.

**Figure 3 sensors-25-00841-f003:**
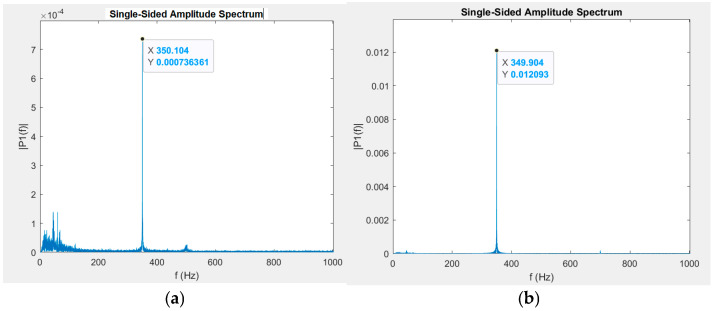
Single-sided amplitude spectrums with an applied AC magnetic field at 350 Hz with amplitudes of 274 nT (**a**) and 2740 nT (**b**) from a Metglas^®^ ribbon sample.

**Figure 4 sensors-25-00841-f004:**
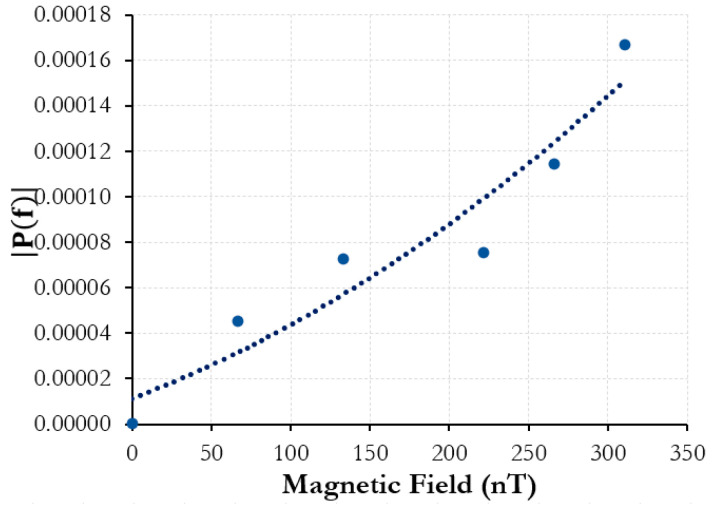
Vitrovac^®^ ribbon fiber sensor: 100 Hz amplitude spectrum intensity vs. 100 Hz AC magnetic-field amplitude.

**Figure 5 sensors-25-00841-f005:**
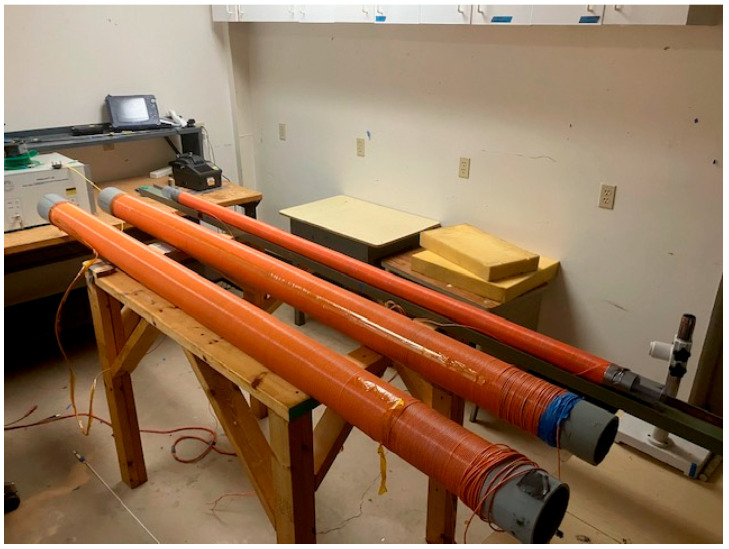
Three air-core solenoids containing Vitrovac^®^ ribbon.

**Figure 6 sensors-25-00841-f006:**
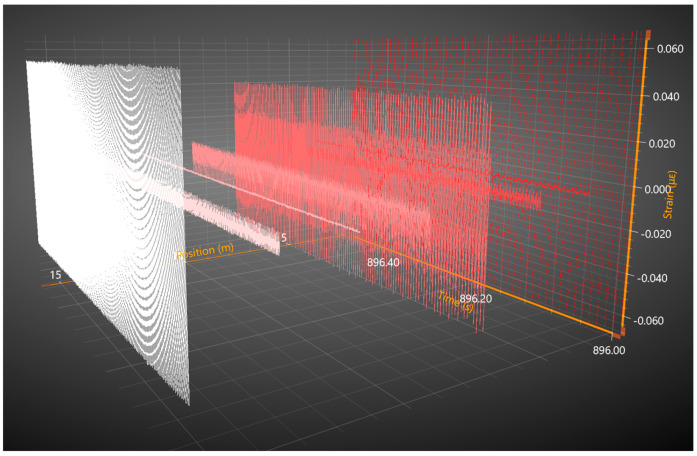
Software capture of 3-dimensional matrix showing position, strain, and time for a half-second interval along 16 m of the fiber and 8 broadband FBG pairs.

**Figure 7 sensors-25-00841-f007:**
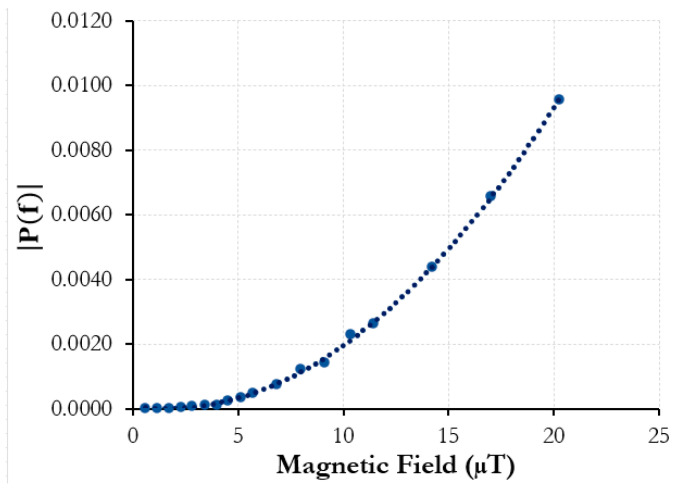
Metglas^®^ ribbon fiber sensor: 200 Hz amplitude spectrum intensity vs. 100 Hz AC magnetic field amplitude.

**Figure 8 sensors-25-00841-f008:**
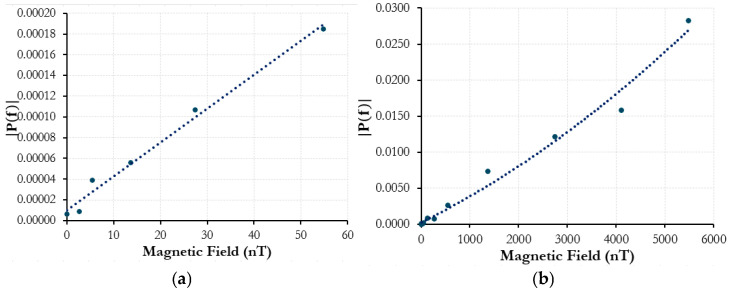
Metglas^®^ ribbon fiber sensor: 350 Hz amplitude spectrum intensity vs. 350 Hz AC magnetic-field amplitude. (**a**) up to 60 nT and; (**b**) up to 6000 nT.

## Data Availability

The raw data supporting the conclusions of this article will be made available by the authors on request.

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
