# Peer review of "Acoustic Sensing Fiber Coupled with Highly Magnetostrictive Ribbon for Small-Scale Magnetic-Field Detection"

_sensors, 2025, doi:10.3390/s25030841_

Round 1
Reviewer 1 Report
Comments and Suggestions for Authors
The manuscript "Acoustic Sensing Fiber Coupled with Highly Magnetostrictive Ribbon for Small Scale Magnetic Field Detection" proposes a method for measuring weak magnetic fields (less than 3 nT). The method is based on the conversion of magnetostrictive oscillations of an amorphous ribbon caused by an alternating magnetic field into an optical signal using an optical fiber and subsequent analysis of this signal.
The manuscript will be of interest to readers of Sensors, since it discusses the problem of measuring weak magnetic fields, which is being studied by many research groups.
However, in the opinion of the reviewer, the manuscript needs revision. Its discussion can be continued after considering the following comments.
1) The Introduction needs to be supplemented. In particular, the Introduction does not discuss weak magnetic field sensors based on the magnetoimpedance effect (MI), which, by the way, is observed in amorphous alloys based on cobalt and iron. In particular, the use of MI sensors for magnetocardiography and magnetoencephalography was previously discussed (see articles by Professor T. Uchiyama, for example, 10.1016/j.jmmm.2020.167148 and 10.1109/TMAG.2019.2895399). Sensors also publishes many articles devoted to the magnetoimpedance effect.
2) The authors need to describe the notations of physical quantities used in the text and figures. For example, it is unclear how P1 (Fig. 3) differs from P (Fig. 4).
3) In the notation P(f), f is the unit of measurement? What is this unit of measurement?
4) Line 110: "…ultra-high sensitivity at less than 0.25 nε". What is ε?
5) Line 111: The authors wrote that the width of the ribbons is 25 mm, and below (lines 113 and 114) they repeat this again. This seems redundant.
6) Fig.6: The labels of some axes are poorly visible.
7) The name of the Vitrovac 7600 T70 alloy and the composition of Co66Fe4Mo2Si16B12 are inconsistent. Vitrovac 7600 T70 is an Fe-based alloy, not a Co-based alloy. On the other hand, the authors found that the response of the Vitrovac ribbons is significantly lower than that of the Metglass ribbons. This suggests that the Vitrovac ribbons have low magnetostriction, which is typical for amorphous Co-based alloys. The authors need to find out where the error is - in the composition or in the name of the alloy.
8) It is necessary to specify the magnetostriction value of the ribbons used.
9) The experiment raises many serious questions. The solenoid field is non-uniform near its ends. The length of the ribbon is equal to the length of the solenoid. Therefore, the ribbon ends are in the region of the non-uniform field and additional stretching forces act on the ribbon, proportional to its magnetic permeability and the field gradient. That is, the deformation of the ribbon, in addition to magnetostriction, may also be caused by this effect. How did the authors take this effect into account?
10) The authors did not compensate for the Earth's magnetic field, even though they are investigating the possibility of measuring very weak magnetic fields. The reviewer believes that this is a very significant drawback of the experiment, which complicates the interpretation of its results. The authors could have compensated at least the longitudinal component of the Earth's magnetic field by applying a corresponding DC to the solenoid.
11) The dimensions of the device are very large, the optical fiber is not fixed to the tape. All this, in the opinion of the reviewer, limits the practical application of the sensor. How do the authors see a solution to these problems?
Author Response
MDPI Sensors Revisions
Editor and Reviewer comments:
Reviewer 1:
1) The Introduction needs to be supplemented. In particular, the Introduction does not discuss weak magnetic field sensors based on the magnetoimpedance effect (MI), which, by the way, is observed in amorphous alloys based on cobalt and iron. In particular, the use of MI sensors for magnetocardiography and magnetoencephalography was previously discussed (see articles by Professor T. Uchiyama, for example, 10.1016/j.jmmm.2020.167148 and 10.1109/TMAG.2019.2895399). Sensors also publishes many articles devoted to the magnetoimpedance effect.
Response: The authors appreciate the comment and the suggested papers. MI needs to be addressed for adequately discussing the relevance of the current magnet field sensors in the biotechnology field. The following was added to further provide context for relevant literature and MI (lines 52-57):
- “Magnetoimpedance (MI) based sensors are another method for highly sensitive magnetic field measurements and have more recently attracted interest in health monitoring with MCG and MEG. Able to resolve field amplitudes in the pico-Tesla regime as well as not having to rely on cryotemperatures to operate, the technique has good potential. However, MI is still based on point measurements and does not have distributed sensing capabilities [5].”
This comes from one of the papers suggested by the reviewer.
2) The authors need to describe the notations of physical quantities used in the text and figures. For example, it is unclear how P1 (Fig. 3) differs from P (Fig. 4).
3) In the notation P(f), f is the unit of measurement? What is this unit of measurement?
4) Line 110: "…ultra-high sensitivity at less than 0.25 nε". What is ε?
Response (2-4): The authors thank the reviewer for the suggestions ensuring units and labels are clarified so the following has been added (lines183-199 ):
- “P1(f) indicates the FFT derived amplitude intensity for a given sinusoidal constituent corresponding with the denoted frequency spectrum along the x-axis... The peaks are selected from these generated frequency spectrum intensity plots, like Figure 3.”
Strain is represented by ε and has been denoted (line 123) and the P(f) discrepancy has been addressed (line 183-201). Additional clarifications have been made across the manuscript to accommodate.
5) Line 111: The authors wrote that the width of the ribbons is 25 mm, and below (lines 113 and 114) they repeat this again. This seems redundant.
Response: The authors appreciate the catch and have fixed the redundancy (line 126-127).
6) Fig.6: The labels of some axes are poorly visible.
Response: The authors agree that figure 6 could have better visibility but do not currently have access to the software to adjust the capture (will attempt to resolve this before the final edits are approved). The 3 axes are individually denoted in the image description to compensate.
7) The name of the Vitrovac 7600 T70 alloy and the composition of Co66Fe4Mo2Si16B12 are inconsistent. Vitrovac 7600 T70 is an Fe-based alloy, not a Co-based alloy. On the other hand, the authors found that the response of the Vitrovac ribbons is significantly lower than that of the Metglass ribbons. This suggests that the Vitrovac ribbons have low magnetostriction, which is typical for amorphous Co-based alloys. The authors need to find out where the error is - in the composition or in the name of the alloy.
8) It is necessary to specify the magnetostriction value of the ribbons used.
Response (7-8): The authors appreciate catching the error and also agree the magnetostriction values should be included. The reviewer is absolutely correct that Vitrovac is an Fe-based alloy and the composition has been corrected (must have been a mix up in recording what was actually used, Vitro 7600 T7 is correct but the initial composition listed was not). The following has been added to address these issues (lines 126-129):
- “and Vitrovac® (7600 T70) Fe65Co18Mo3Si0.8B15.5C0.4. The Metglas® measured .025mm in thickness while the Vitrovac® was .021mm thick. Vitrovac® 7600 T70 has a saturation magnetostriction coefficient of around 47 ppm and Metglas® 2605 SC of 27 ppm.”
Even though Vitrovac has a higher saturation magnetostriction coefficient than Metglas, the slope of the magnetostriction curve, strain vs magnetic field, may still be higher than that of Vitrovac where we were doing our measurements (dependent on AC field amplitude as well static field bias such as from the earth). Additionally, the ribbon thickness is not the same which could affect the acoustic coupling and thus resultant strain response. From other unpublished work in our group, Vitrovac does still appear to have a higher strain response at higher fields >100uT which agrees with the coefficient values. More needs to be done to determine exactly why this is the case (which we are working on), but the authors believe it is beyond the scope of this specific communications article.
9) The experiment raises many serious questions. The solenoid field is non-uniform near its ends. The length of the ribbon is equal to the length of the solenoid. Therefore, the ribbon ends are in the region of the non-uniform field and additional stretching forces act on the ribbon, proportional to its magnetic permeability and the field gradient. That is, the deformation of the ribbon, in addition to magnetostriction, may also be caused by this effect. How did the authors take this effect into account?
Response: The authors agree that the end effects of the solenoid should be taken into account. It was during experimentation but the figure demonstrating the setup was improperly labeled in its dimensions. The solenoids used are actually 2.2 meters in length and Figure 2 has been relabeled. (you can see in the diagram the 2m ribbon does not reach the end of the cylinder/solenoid). 10 cm on either side of leeway room (5%) of the ribbon length should account for most of these distortions. We apologize for the confusion and error and should have specified.
10) The authors did not compensate for the Earth's magnetic field, even though they are investigating the possibility of measuring very weak magnetic fields. The reviewer believes that this is a very significant drawback of the experiment, which complicates the interpretation of its results. The authors could have compensated at least the longitudinal component of the Earth's magnetic field by applying a corresponding DC to the solenoid.
Response: The authors agree that Earth’s magnetic field should be addressed and thank the reviewer for the comment. In this case, since we are measuring just AC fields, Earth’s magnetic field only shifts where the measurement is being acquired along the magnetostriction curve for the given material and being on the order of tens of microTesla, the resultant strain response is not changed significantly (aside from the frequency strain response shifting from doubled to driving which was accounted for and noted). We wanted to show a potential sensor without the need for adding shielding or a DC magnetic field offset. In actual applications the sensor would have to be calibrated with this in mind. This is briefly noted in lines 217-223 but is not further elaborated due to the scope of the communication manuscript and to remain concise.
11) The dimensions of the device are very large, the optical fiber is not fixed to the tape. All this, in the opinion of the reviewer, limits the practical application of the sensor. How do the authors see a solution to these problems?
Response: The authors appreciate the question and agree this is a potential drawback to the sensor. Additional unpublished testing was done with Scotch tape ensuring the fiber does not move and following text was included (lines 298-303):
- “Vibration from the metal itself or environmental movement could also cause the acoustic sensing fiber to shift its position from its centered alignment along the ribbon which could in turn alter the detected strain response by the fiber. However, the fiber was later taped down using Scotch® tape to remove the possibility of the fiber position changing and no significant difference was shown in sensitivity, but consistency of repeated tests was improved.”
As the tape only improved the sensor, to move forward with experimentation or designing a cable, this would not be a drawback in the future. The sensor configuration itself is very compact only relying on the fiber and ribbon which is 2.5 cm in width which could be further reduced. The solenoid was just for testing field strengths. While the spatial resolution is two meters, the sensing configuration could be coiled or folded back on itself for a small profile and the following is added to address this (lines 316-318):
- “In applications where a 2m gauge length and sensing resolution is excessive, the fiber and ribbon could be coiled to reduce the physical footprint of the sensor with further testing to characterize this.”
Reviewer 2 Report
Comments and Suggestions for Authors
The paper proposed methods of magnetic characterization capable of
distributed and high sensitivity field measurements. The strain response
from the highly magnetostrictive alloys and an applied AC magnetic field are
investigated. A minimal detectable field amplitude of 60 nT was achieved.
Authors should address the below questions:
1. Authors have to demonstrate the advantages of using DAS for magnetic
measurement in the introduction section. They mentioned in line 45:
“ However, devices and sensors in the biomedical field used for real time
magnetic sensing diagnostics are expensive...” However, DAS is also
expensive. I suggest that more support references have to be included.
2. In line 66 : “However, vertical resolution with NMR is limited and
obtaining enough information can be slow and cumbersome.” Reference
should be added.
3. Authors mentioned “small scale”. Please provide experimental data
regarding the sensor detection range of the magnetic field.
4. The advantages of using DAS compared with point measurement was not
experimentally demonstrated. Authors also have to show the location
accuracy of the proposed magnetic sensor.
5. What is the loss of the acoustic optical fiber? I would also like to know the
maximum sensing distance of the proposed sensor?
6. More measurement results should be provided to demonstrate the
distributed sensing performance with more than two magnetic fields at
different location.
Author Response
MDPI Sensors Revisions
Editor and Reviewer comments:
Reviewer #2:
1). Authors have to demonstrate the advantages of using DAS for magnetic measurement in the introduction section. They mentioned in line 45: “However, devices and sensors in the biomedical field used for real time magnetic sensing diagnostics are expensive...” However, DAS is also expensive. I suggest that more support references have to be included.
2). In line 66 : “However, vertical resolution with NMR is limited and obtaining enough information can be slow and cumbersome.” Reference should be added.
Response (1-2): The authors appreciate the comment and agree that more information should be added. Five additional references have been included and cited accordingly in the updated manuscript. The following text has been added using some of these new sources to address both NMR and DAS (lines:71-75 and lines 82-88):
- “Surface-NMRs conversely can take bulk measurements in real time and are not limited by the scarcity of legacy cores. In some recent studies, NMR has shown newfound potential with advancements in sensing information [8]. However, vertical resolution with NMR is usually limited and obtaining enough information can be slow and cumbersome [6][8].”
- “Distributed acoustic sensing (DAS) in particular has been reliably used as a measuring device for surveillance and structural health monitoring. DAS optical fiber can distinguish acoustic signals across several kilometers along its length [10][11]. The high sampling rate (thousands of Hz) allows for a large amount of information to be obtained about individual vibrations anywhere along the length of the fiber including the phase, frequency, and amplitude. Used in vehicle monitoring, seismic activity detection, and various other applications, the distributed nature of the technology and high sensitivity lends itself to being one of the most promising methods for detection of acoustic signals [11].”
3). Authors mentioned “small scale”. Please provide experimental data regarding the sensor detection range of the magnetic field.
Response: The authors thank the reviewer for the note and also think that small scale is vague. However, in the abstract and conclusion the minimum sensitivity of 3nT was stated (line 20 and 331 respectively). Figure 8 shows the FFT amplitude intensity peaks plotted against magnetic field demonstrating that at 2.74nT the corresponding strain response peak was able to be recorded above the noise floor. Figure 7 shows a much larger sensitivity range for the Metglas reaching fields up to 20uT. While the complete detection range for the sensor configuration is in reality larger than this being able to detect a wider range of frequencies and higher amplitudes than 20uT, for the scope of the communications manuscript and conciseness only the data provided is included.
4). The advantages of using DAS compared with point measurement was not experimentally demonstrated. Authors also have to show the location accuracy of the proposed magnetic sensor.
5). What is the loss of the acoustic optical fiber? I would also like to know the maximum sensing distance of the proposed sensor?
6). More measurement results should be provided to demonstrate the distributed sensing performance with more than two magnetic fields at different location.
Response (4-6): The authors agree that more could be included both experimentally and in terms of literature to highlight the capabilities of DAS and information about the sensing fiber. We appreciate the suggestions and have edited the manuscript to better accommodate this. The following text was added to provide clarity (lines 245-253):
- “All three signals were unique, and the spectrum peaks corresponded to their respective solenoid driving frequency with components of both the doubled response and original sinusoid. The FBG pair in the first solenoid is 4 meters away from the FBG pair in the second which is then 6 meters away from the pair in the third and final solenoid. Even though the experiment demonstrates a sensing spatial resolution of 2 meters along only 16 meters of fiber, the used DAS system is capable of distributed measurements along tens of kilometers of fiber with minimal loss. General DAS fibers typically exhibit .4 db per kilometer of loss and have been shown to still resolve over 100 kilometers worth of fiber [11].”
Three independently resolvable magnetic fields were demonstrated with the sensor configuration at a spatial resolution of 2 meters (Figure 6). While this does not represent the full capability of DAS, the group has limited funding and procuring data to adequately characterize the full potential would be costly and time consuming. As a stand in, information from another study was included to briefly address it. However, this is something that we would like to do in the future.
Round 2
Reviewer 1 Report
Comments and Suggestions for Authors
The authors have taken into account all the comments of the Reviewer and have made the appropriate changes to the manuscript. The Reviewer hopes that the authors will improve Fig. 6.